# Enhancing News Article Classification in Low-Resource Languages: A Supervised Contrastive-Masked Pretraining Framework

## Abstract

News article classification in low-resource languages often faces significant challenges due to limited availability of labeled data and insufficient exposure of large language models (LLMs) to these languages during pretraining. To address these issues, we introduce Supervised Contrastive Masked Pretraining (SCMP), a novel approach designed to enhance the performance of LLMs in low-resource settings. SCMP integrates supervised contrastive learning with masked language modeling (MLM) during pretraining, effectively leveraging limited labeled data to improve the model's ability to distinguish between classes while capturing meaningful semantic representations. Additionally, during fine-tuning, we introduce a joint loss function that combines classification and MLM objectives, ensuring that the model retains essential contextual knowledge while adapting efficiently to downstream tasks. Beyond improving accuracy, SCMP reduces dependence on large labeled corpora, making it a practical solution for large-scale or dynamic multilingual news classification pipelines. Experiments on nine Indian and seven African languages demonstrate that SCMP consistently outperforms standard fine-tuning approaches. Our findings suggest that incorporating supervised contrastive objectives into masked pretraining, coupled with a joint fine-tuning strategy, offers a resource-effective framework for advancing LLM performance in low-resource linguistic environments. Code will be released upon acceptance.

## 1 Introduction

Pretraining has become a fundamental technique in natural language processing (NLP), with masked language modeling (MLM) playing a crucial role in self-supervised learning. *BERT* (Devlin, 2018) introduced the MLM objective, enabling models to learn deep contextualized representations by predicting masked tokens in a bidirectional context. Later models, such as *RoBERTa* (Liu et al., 2021), improved upon *BERT* by removing the next sentence prediction (NSP) objective, dynamically changing the masking patterns across training epochs, leveraging much larger batch sizes, and training on significantly more data. These refinements led to more robust representations and enhanced downstream task performance. However, standard MLM-based models, including RoBERTa, often require substantial computational resources, making them less practical for low-resource language applications.

To address the computational overhead of large-scale transformers, during both training and inference, models like *DistilBERT* (Sanh et al., 2019) were introduced, offering a compact alternative while maintaining much of the performance of *BERT*. *DistilBERT uncased*, in particular, has demonstrated effectiveness in resource-constrained environments by reducing model size while preserving essential language understanding capabilities. Despite these advancements, existing pretraining methods often fail to explicitly enhance feature discrimination, which is critical for low-resource language classification.

Contrastive learning is a powerful technique for representation learning that explicitly encourages similar samples to have closer embeddings while pushing dissimilar ones apart. In computer vision, SimCLR (Chen et al., 2020) demonstrated that extensive data augmentations combined with a contrastive loss can yield robust image representations in a self-supervised setting. Separately, supervised contrastive learning (Khosla

et al., 2020) leverages label information to enhance representation quality by better distinguishing semantically related samples from unrelated ones. In natural language processing, similar contrastive objectives have been applied to learn sentence-level representations, enabling models to capture fine-grained semantic similarities between sentences (Zhang et al., 2022). Despite these, the integration of supervised contrastive objectives with masked language modeling (MLM) remains relatively underexplored—particularly for downstream tasks in low-resource languages, where efficient representation learning is critical.

Low-resource languages present unique challenges for standard pretraining and fine-tuning pipelines. Multilingual models like *multilingual BERT* (Devlin, 2018) and *XLM-R*(Conneau, 2019) have made significant strides by transferring knowledge across languages. Nonetheless, these models may not fully capture language-specific nuances essential for tasks such as news classification. Recent studies have investigated strategies to inject additional supervision during pretraining to better serve low-resource languages; yet, none have explicitly combined supervised contrastive learning with MLM to exploit both global semantic relationships and local token dependencies.

Finally, the design of joint objectives during fine-tuning has attracted attention as a means to mitigate catastrophic forgetting and preserve the benefits of pretraining. Prior work in *multi-task learning* (Liebel & Körner, 2018) has shown that maintaining auxiliary objectives can lead to improved downstream performance. In our approach, we extend this idea by continuing to optimize an MLM loss alongside the classification loss during fine-tuning. This joint loss formulation aims to retain contextual knowledge acquired during pretraining while adapting the model to the classification task, a strategy that is particularly beneficial when labeled data is limited.

Although both masked language modeling (MLM) and supervised contrastive learning (SCL) have been explored independently in prior work, the specific integration we introduce in Supervised Contrastive–Masked Pretraining (SCMP) represents a novel contribution. Existing literature either applies these objectives in isolation or combines them only within the downstream fine-tuning stage, leaving a conceptual and empirical gap that SCMP addresses.

Domain-adaptive pretraining (Gururangan et al., 2020) demonstrates the value of continued MLM on in-domain text but does not incorporate supervised contrastive objectives during pretraining. Conversely, Gunel et al. (2021) applies supervised contrastive loss only during fine-tuning—paired with cross-entropy—but does not explore its combination with MLM at the pretraining stage. In contrast, SCMP introduces a unified pretraining objective that jointly optimizes supervised contrastive loss and MLM loss, a combination that has not been evaluated in previous studies.

This design fills a key conceptual gap: prior methods treat contrastive learning and MLM as separate mechanisms, with contrastive learning typically shaping sentence-level representations during fine-tuning while MLM focuses on token-level reconstruction during pretraining. SCMP is the first framework to integrate these complementary signals directly within the pretraining phase, enabling the model to learn both discriminative class-aware embeddings and robust contextualized token representations before exposure to the downstream classifier.

SCMP also addresses an empirical gap, particularly in low-resource settings. While earlier studies report improvements in high-resource languages, the impact of a unified contrastive–masking objective on low-resource news classification has not been systematically investigated. Our work demonstrates that this integration yields substantial benefits across sixteen Indic and African languages, highlighting the practical advantages of SCMP for real-world multilingual applications.

## 2 SCMP Pretraining and Joint Fine Tuning Framework

### 2.1 Contrastive–MLM Loss $\mathcal{L}_{\mathrm{CML}}$

In this section, we describe our formulation of the loss function for pretraining, which combines a contrastive term with a masked token prediction objective. For clarity, we begin by defining our dataset and notation.

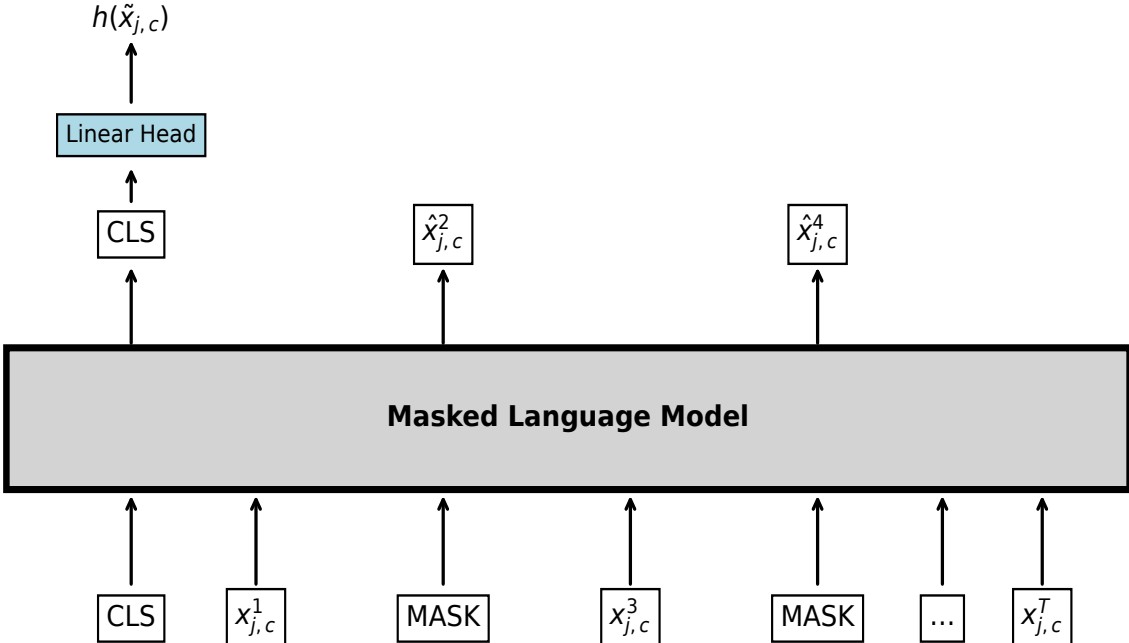

Figure 1: Overview of the proposed Supervised Contrastive-Masked Pretraining (SCMP) framework. Given a sample $x_{j,c}$ from class $c$, the input sequence consists of a CLS token, observed tokens $x_{j,c}^1, x_{j,c}^3, x_{j,c}^5$, and masked tokens. The Masked Language Model processes the sequence, outputting predicted logits $\hat{x}_{j,c}^2$ and $\hat{x}_{j,c}^4$ for the masked tokens $x_{j,c}^2$ and $x_{j,c}^4$. The CLS token, which captures the global semantic representation of the entire sequence, is passed through a linear head to obtain the embedding $h(\tilde{\boldsymbol{x}}_{j,c})$.

Let

$$\mathcal{C} = \{1, 2, \ldots, n\} \tag{1}$$

denote the set of classes in our dataset. For each class $c \in \mathcal{C}$, let $\mathcal{D}_c$ represent the set of samples corresponding to class $c$. From each $\mathcal{D}_c$, we randomly select two samples, denoted as follows:

$$\boldsymbol{x}_{1,c} \quad \text{and} \quad \boldsymbol{x}_{2,c}. \tag{2}$$

For each sample $\boldsymbol{x}_{j,c}$ (with $j \in \{1, 2\}$), we define the following notation:

- $x_{j,c}^i$ represents the $i$th token of the sample $\boldsymbol{x}_{j,c}$.

- $\boldsymbol{m}_{j,c}$ is the set of indices corresponding to tokens that are masked during training.

- $\tilde{\boldsymbol{x}}_{j,c}$ denotes the masked version of $\boldsymbol{x}_{j,c}$.

The contrastive MLM loss (CML) consists of two components: (1) a contrastive loss (CL) that aligns representations of samples from the same class and (2) masked language modeling loss (ML), which drives the model to reconstruct the masked tokens. Formally, the loss is defined as

$$\mathcal{L}_{\text{CML}} = \underbrace{-\sum_{c=1}^{n} \log \frac{e^{-\|h(\tilde{\boldsymbol{x}}_{1,c}) - h(\tilde{\boldsymbol{x}}_{2,c})\|^2}}{\sum_{c'=1}^{n} e^{-\|h(\tilde{\boldsymbol{x}}_{1,c}) - h(\tilde{\boldsymbol{x}}_{2,c'})\|^2}}}_{\text{Contrastive loss on masked inputs(CL)}} + \underbrace{-\frac{1}{2n} \sum_{c=1}^{n} \sum_{j=1}^{2} \frac{1}{|\boldsymbol{m}_{j,c}|} \sum_{i \in \boldsymbol{m}_{j,c}} \log p(x_{j,c}^i \mid \tilde{\boldsymbol{x}}_{j,c})}_{\text{MLM loss(ML)}}. \tag{3}$$

Here, $h(\cdot)$ denotes the linearly transformed CLS embedding (Devlin, 2018) from the masked language model, and $p(x_{j,c}^i \mid \tilde{\boldsymbol{x}}_{j,c})$ is the probability assigned by the model to the masked token $x_{j,c}^i$ given its masked context $\tilde{\boldsymbol{x}}_{j,c}$.

**Tensorized Contrastive Loss**

To compute the Contrastive term (CL) in equation 3 over a batch, we arrange the masked inputs into two tensors,

$$\mathbf{X}_1 = \{\tilde{\mathbf{x}}_{1,c} : c \in \mathcal{C}\}, \qquad \mathbf{X}_2 = \{\tilde{\mathbf{x}}_{2,c} : c \in \mathcal{C}\},$$

each containing one masked sample per class. For each anchor embedding $h(\tilde{\mathbf{x}}_{1,c})$, the contrastive term in equation 3 compares it against all candidate positives and negatives $\{h(\tilde{\mathbf{x}}_{2,c'})\}_{c'=1}^n$, treating $c' = c$ as the positive and all other classes as negatives. This set of $n$ pairwise comparisons can be written compactly by forming the matrix of pairwise similarities $\mathbf{Z} \in \mathbb{R}^{n \times n}$,

$$\mathbf{Z} = -\Big(\mathbf{1}_n \, \mathrm{vec}\big(h(\mathbf{X}_1)\big) - \big(\mathbf{1}_{1 \times n} \otimes h(\mathbf{X}_2)\big)\Big)^{\circ 2} (I_n \otimes \mathbf{1}_d), \tag{4}$$

which yields

$$Z_{c,c'} = -\| h(\tilde{\mathbf{x}}_{1,c}) - h(\tilde{\mathbf{x}}_{2,c'}) \|^2.$$

Applying a row-wise softmax to $\mathbf{Z}$ yields, for each anchor $h(\tilde{\mathbf{x}}_{1,c})$, the normalized similarity distribution over all classes. This operation reproduces exactly the denominator of CL component in equation 3, and extracting the diagonal entry of softmax($\mathbf{Z}$) selects the probability assigned to the correct positive pair for each class. Thus, the contrastive term admits the compact tensorized expression

$$\mathcal{L}_{\mathrm{CL}} = -\sum_{i=1}^{n} \log\big(\big[\mathrm{softmax}(\mathbf{Z})\big]_{ii}\big), \tag{5}$$

which is algebraically equivalent to the CL term in Eq. equation 3, but enables a tensorized implementation. As $n$ increases, the computation remains inside optimized CUDA kernels—avoiding Python-level loops and thereby scaling more efficiently (see Appendix C).

**Notation.**

- $\mathbf{1}_n \in \mathbb{R}^{n \times 1}$ is a column vector of ones of length $n$.

- $\mathbf{1}_{1 \times n} \in \mathbb{R}^{1 \times n}$ is a row vector of ones.

- $\otimes$ denotes the Kronecker product.

- $I_n \in \mathbb{R}^{n \times n}$ is the $n \times n$ identity matrix.

- $\mathbf{1}_d \in \mathbb{R}^{d \times 1}$ is a column vector of ones of length $d$, where $d$ is the embedding dimension.

- The operator $(\cdot)^{\circ 2}$ applies an element-wise square.

- The subscript $[\cdot]_{ii}$ selects the diagonal element (corresponding to class $i$) from the softmax output, which is applied row-wise (i.e., computed independently for each row).

- $\mathrm{vec}(h(\boldsymbol{X}_1)) \in \mathbb{R}^{1 \times (n\,d)}$ is the *row-major vectorization* of $h(\boldsymbol{X}_1) \in \mathbb{R}^{n \times d}$ (*Golub&Van Loan*, 2013).

## 2.2 Joint Fine Tuning Strategy

To adapt the SCMP pretrained model for classification, we replace the pretraining linear head with a classification head, which projects the CLS (classification) embedding (Devlin, 2018) into class logits. Given a labeled mini-batch $\mathcal{B} = \{(\boldsymbol{x}_b, y_b)\}_{b=1}^B$ with masked indices $\boldsymbol{m}_b$ for each sample $\boldsymbol{x}_b$, the joint fine-tuning loss (FT) is defined as:

$$\mathcal{L}_{\text{FT}} = \underbrace{-\frac{1}{B}\sum_{b=1}^{B}\frac{1}{|\boldsymbol{m}_b|}\sum_{i\in\boldsymbol{m}_b}\log p(x_b^i \mid \tilde{\boldsymbol{x}}_b)}_{\text{MLM loss}} + \underbrace{-\frac{1}{B}\sum_{b=1}^{B}\log p(y_b \mid \text{softmax}(\mathbf{W}h(\boldsymbol{x}_b) + \boldsymbol{\beta}))}_{\text{Classification loss}}. \tag{6}$$

where:

- $p(x_b^i \mid \tilde{\boldsymbol{x}}_b)$ is the probability of predicting the masked token $x_b^i$ from its context.

- $h(\boldsymbol{x}_b)$ denotes the CLS embedding capturing the input's global semantics.

- The classification head is parameterized by $\mathbf{W} \in \mathbb{R}^{n\times d}$ and $\boldsymbol{\beta} \in \mathbb{R}^n$, with $n$ classes and embedding dimension $d$.

- $p(y_b \mid \cdot)$ represents the predicted probability of the true class.

By jointly optimizing masked language modeling and classification losses, our approach retains contextual knowledge from pretraining while effectively adapting the model to classification tasks.

## 3 Results and Analysis

### 3.1 Dataset Overview and Performance

Our experiments leverage two distinct news classification datasets to evaluate our method across a diverse set of low-resource languages.

For the nine Indian languages, we use the ***IndicNLP News Article Classification Dataset*** (Kunchukuttan et al., 2020). This dataset, curated from various online news sources, was created using a semi-automated labeling process where article categories were inferred from URL components. The corpus is balanced across classes to ensure robust model training.

To broaden the linguistic scope of our evaluation, we also incorporate data for seven African languages from the **MasakhaNEWS** dataset (Adelani et al., 2023). The articles for this dataset were sourced from the BBC and VOA and annotated by native-speaking volunteers from the Masakhane community. The curation involved a two-stage active learning process to improve efficiency: annotators first manually labeled an initial set of articles and then corrected the predictions of a classifier on the remaining data, with final labels assigned by majority vote.

**Table 1** provides a detailed breakdown of the number of classes and the distribution of training and test samples for all sixteen languages used in our study.

To evaluate the effectiveness of the proposed loss functions introduced in this paper, we trained the *DistilBERT-multilingual-cased* (Sanh et al., 2019) model using the contrastive MLM loss (Equation 3). Importantly, for each language, the train/test split was performed *before* any pretraining, and all supervised contrastive MLM losses were derived solely from the training portion to avoid any information leakage. After pretraining, we replaced the linear head with a classification head on the CLS embedding and fine-tuned the model using the joint loss (Equation 6). To ensure a fair comparison across languages, we trained separate models for each language in the dataset. Additionally, we directly fine-tuned the base model without contrastive masked pretraining or the masked language modeling objective.

Table 1: Number of classes, training samples, and test samples for each language in the IndicNLP News Article Classification Dataset, plus 7 African languages from MasakhaNEWS. Each sample represents a news article.

| Language | Number of Classes | Train Samples | Test Samples |
|---|---|---|---|
| *Indic languages* | | | |
| Bengali | 2 | 11,199 | 1,399 |
| Gujarati | 3 | 1,631 | 203 |
| Kannada | 3 | 23,999 | 2,999 |
| Malayalam | 4 | 4,799 | 599 |
| Marathi | 3 | 3,814 | 477 |
| Oriya | 4 | 23,999 | 2,999 |
| Punjabi | 4 | 2,495 | 311 |
| Tamil | 3 | 9,359 | 1,169 |
| Telugu | 3 | 19,199 | 2,399 |
| *African languages* | | | |
| Hausa | 7 | 2,219 | 637 |
| Igbo | 6 | 1,356 | 390 |
| Kiswahili | 7 | 1,658 | 476 |
| Luganda | 5 | 771 | 223 |
| Lingala | 4 | 608 | 175 |
| Pidgin | 5 | 1,060 | 305 |
| Yorùbá | 5 | 1,433 | 411 |

Table 2 compares the performance of direct fine-tuning against our SCMP with joint fine-tuning across sixteen languages. For the Indic languages, our method provides consistent and significant improvements. In high-performing languages like Gujarati, Kannada, and Telugu, accuracy increases from around 0.96–0.97 to 0.99. The benefits are even more pronounced for languages with lower baselines; for example, Malayalam's accuracy rises substantially from 0.89 to 0.94, and Marathi improves from 0.95 to 0.99.

This trend of improvement extends to most of the African languages. The largest gains are seen in Igbo, where accuracy jumps from 0.74 to 0.83, and Yorùbá, which increases from 0.80 to 0.89. Modest-to-strong gains are also observed for Hausa, Kiswahili, and Luganda. There are two exceptions: performance on Pidgin remains unchanged, and Lingala shows a marginal decrease in performance with our method.

Overall, these results confirm that integrating a supervised contrastive signal during pretraining, coupled with a joint fine-tuning objective, enhances model generalization across a diverse set of languages and is particularly effective for improving performance in lower-resource settings. Beyond this comparison against direct fine-tuning, we also benchmark SCMP against contrastive and domain-adaptive baselines such as SimCSE (Gao et al., 2021), LaBSE (Feng et al., 2022), and DAPT (Gururangan et al., 2020); the language-wise results in Appendix B show that SCMP consistently matches or outperforms these alternatives across all sixteen languages.

To analyze the data efficiency of our framework, we conducted experiments using subsets of the training data. Table 3 presents a spotlight analysis of weighted F1-scores at 20%, 40%, and 60% label budgets for six representative languages, categorized by their baseline strength.

The results clearly demonstrate that the SCMP framework is significantly more data-efficient, especially for languages with weaker baselines. For **Igbo**, our method using only 20% of the labeled data (0.77 F1) outperforms direct fine-tuning with 60% of the data (0.76 F1). Similarly, for **Oriya**, SCMP with a 20% budget achieves a 0.70 F1-score, surpassing the baseline's performance even at the 40% budget (0.68).

Even for languages with strong baselines, such as **Kannada** and **Telugu**, our approach consistently yields higher performance with less data. With just 20% of the labels, SCMP achieves F1-scores of 0.98 and 0.99 respectively, matching or exceeding the baseline's performance at much larger data budgets. These findings

Table 2: Language-wise comparison of precision, recall, F1-score, and accuracy between Direct Fine-Tuning and SCMP with joint fine-tuning on the IndicNLP News Article Classification Dataset and 7 African languages using *DistilBERT-multilingual-cased* as the base model. Precision, recall, and F1-score are weighted across all classes for each language. Scores are averaged over 5 independent runs.

| Language | Direct Fine-Tuning | | | | SCMP + Joint Fine-Tuning | | | |
|---|---|---|---|---|---|---|---|---|
| | **Precision** | **Recall** | **F1-score** | **Accuracy** | **Precision** | **Recall** | **F1-score** | **Accuracy** |
| *Indic languages* | | | | | | | | |
| Bengali | 0.97 | 0.97 | 0.97 | 0.97 | **0.98** | **0.98** | **0.98** | **0.98** |
| Gujarati | 0.96 | 0.96 | 0.96 | 0.96 | **0.99** | **0.99** | **0.99** | **0.99** |
| Kannada | 0.97 | 0.97 | 0.97 | 0.97 | **0.99** | **0.99** | **0.99** | **0.99** |
| Malayalam | 0.89 | 0.89 | 0.89 | 0.89 | **0.94** | **0.94** | **0.94** | **0.94** |
| Marathi | 0.95 | 0.95 | 0.95 | 0.95 | **0.99** | **0.99** | **0.99** | **0.99** |
| Oriya | 0.72 | 0.72 | 0.72 | 0.72 | **0.74** | **0.74** | **0.74** | **0.74** |
| Punjabi | 0.96 | 0.95 | 0.95 | 0.96 | **0.98** | **0.98** | **0.98** | **0.98** |
| Tamil | 0.95 | 0.95 | 0.95 | 0.95 | **0.98** | **0.98** | **0.98** | **0.98** |
| Telugu | 0.97 | 0.97 | 0.97 | 0.97 | **0.99** | **0.99** | **0.99** | **0.99** |
| *African languages* | | | | | | | | |
| Hausa | 0.85 | 0.85 | 0.85 | 0.85 | **0.87** | **0.86** | **0.86** | **0.86** |
| Igbo | 0.76 | 0.74 | 0.75 | 0.74 | **0.83** | **0.83** | **0.83** | **0.83** |
| Kiswahili | 0.77 | 0.76 | 0.76 | 0.76 | **0.80** | **0.79** | **0.79** | **0.79** |
| Luganda | 0.79 | 0.80 | 0.79 | 0.80 | **0.86** | **0.86** | **0.86** | **0.86** |
| Lingala | **0.90** | **0.90** | **0.90** | **0.90** | 0.89 | 0.89 | 0.89 | 0.89 |
| Pidgin | 0.98 | 0.98 | 0.98 | 0.98 | 0.98 | 0.98 | 0.98 | 0.98 |
| Yorùbá | 0.83 | 0.80 | 0.80 | 0.80 | **0.90** | **0.89** | **0.89** | **0.89** |

Table 3: Spotlight data-efficiency results on six languages: Bengali, Kannada, and Telugu (strongest baselines) and Oriya, Malayalam, Igbo (weakest baselines). We report weighted F1 at 20%, 40%, and 60% label budgets.

| Language | Direct Fine-Tuning | | | SCMP + Joint Fine-Tuning | | |
|---|---|---|---|---|---|---|
| | **20%** | **40%** | **60%** | **20%** | **40%** | **60%** |
| *Strong baselines* | | | | | | |
| Bengali | 0.96 | 0.97 | 0.98 | **0.97** | **0.98** | 0.98 |
| Kannada | 0.97 | 0.98 | 0.97 | **0.98** | **0.99** | **0.99** |
| Telugu | 0.98 | 0.98 | 0.97 | **0.99** | **0.99** | **0.99** |
| *Weak baselines* | | | | | | |
| Oriya | 0.61 | 0.68 | 0.72 | **0.70** | **0.71** | **0.74** |
| Malayalam | 0.82 | 0.88 | 0.90 | **0.88** | 0.88 | 0.90 |
| Igbo | 0.68 | 0.74 | 0.76 | **0.77** | **0.77** | **0.79** |

underscore that our method not only improves peak performance but also offers a more robust learning trajectory in data-scarce scenarios.

## 3.2 Feature Representation: t-SNE Visualization

To assess the impact of Supervised Contrastive-Masked Pretraining (SCMP) on feature representation, we employed t-SNE (t-distributed Stochastic Neighbor Embedding) visualizations using the Potrika Bengali News Classification Dataset (Ahmad et al., 2022). This dataset, with its eight distinct classes, offers a more challenging and diverse classification space, making it particularly well-suited for analyzing how SCMP enhances the discriminability of the learned embeddings. Specifically, we visualize the t-SNE projections

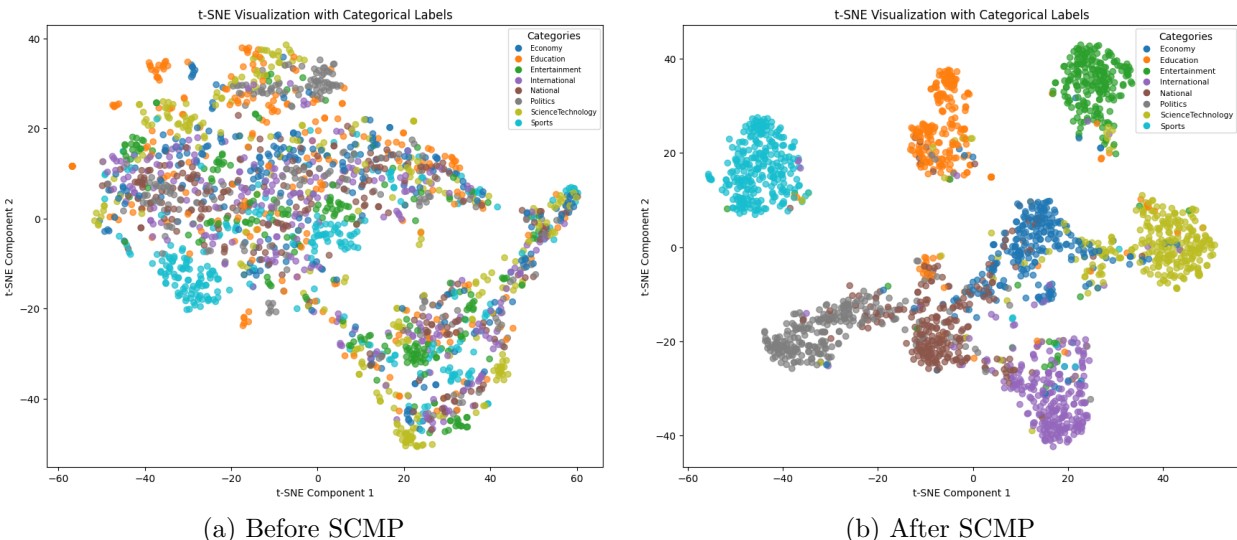

(a) Before SCMP          (b) After SCMP

Figure 2: t-SNE visualization of CLS embeddings. (a) Pretrained *DistilBERT-multilingual-cased* (before SCMP) and (b) after SCMP pretraining. Colors denote class labels. SCMP results in tighter intra-class clusters and clearer inter-class separation.

of the CLS embeddings of news articles from the test set across eight different classes after the model was pretrained on the training samples with the contrastive MLM loss (Equation 3).

**Impact of SCMP.** As shown in Figure 2a, before applying SCMP, the embeddings exhibit significant overlap between classes, indicating weak feature discrimination. In contrast, Figure 2b demonstrates that SCMP improves cluster compactness and enhances inter-class separability. The integration of contrastive learning during masked pretraining helps the model differentiate between classes more effectively while masked language modeling (MLM) preserves contextual semantics.

These qualitative results align with our quantitative improvements in classification accuracy, confirming that SCMP enhances feature separability, ultimately improving the model's generalization performance in a low-resource setting.

### 3.3 Training Dynamics and Convergence

We further examine the training dynamics of SCMP on the Potrika Bengali News Classification Dataset (Ahmad et al., 2022). Figure 3 compares the training and validation accuracy curves between:

1. Direct Fine-Tuning: The base *DistilBERT-multilingual-cased* model fine-tuned on the classification task without SCMP pretraining.

2. Joint Fine-Tuning After SCMP: Fine-tuning with the joint loss objective (Equation 6) after SCMP pertaining with (Equation 3).

**Findings.** In the joint fine-tuning approach, Figure 3a, the model rapidly achieves over 80% validation accuracy in the first epoch. This early boost can be attributed to the SCMP pretraining phase, which learns both discriminative representations across samples and token-level dependencies through MLM. As fine-tuning proceeds, the validation accuracy exhibits a generally upward learning trajectory and stabilizes around 88%, reflecting how the model's ability to preserve semantic representations learned in pretraining while adapting to the classification task contributes to stronger generalization.

Direct fine-tuning, as shown in Figure 3b, starts at a lower validation accuracy and displays a more downward learning curve. Although its validation accuracy initially improves, performance begins to degrade in later

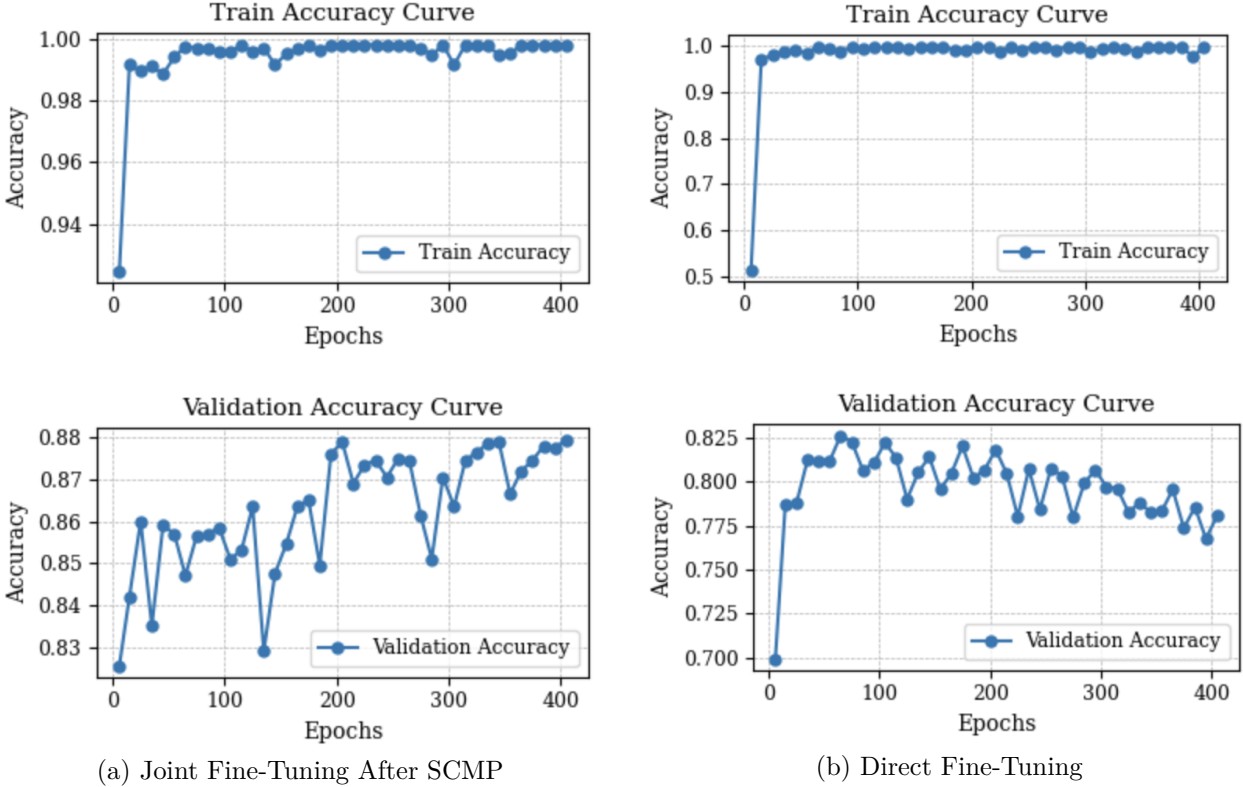

(a) Joint Fine-Tuning After SCMP  (b) Direct Fine-Tuning

Figure 3: **Training and Validation Accuracy Curves.** (a) With joint fine-tuning after SCMP, the validation accuracy exhibits a generally upward learning trajectory and stabilizes around 88%, indicating effective learning and stronger generalization without significant overfitting. (b) In direct fine-tuning, validation accuracy initially increases but starts to decline in later epochs, suggesting overfitting as the model memorizes training data rather than generalizing effectively.

epochs. The absence of an auxiliary MLM objective during fine-tuning appears to leave the model vulnerable to overfitting, as it lacks a mechanism to reinforce contextual understanding. As a result, the model tends to memorize training examples rather than generalize based on semantic relationships, leading to a decline in performance over time.

These observations confirm that SCMP not only improves classification performance but also ensures stable and efficient learning, reducing the risk of overfitting in low-resource settings.

### 3.4 Ablation Study

To better understand the individual contributions of each component in our framework, we conduct an ablation study evaluating three distinct training strategies. First, following the domain adaptation approach proposed by Gururangan et al. (2020), we apply Continued Masked Pretraining (CMP), where the base model is further pretrained on unlabeled in-domain news data using the masked language modeling (MLM) objective. This continued pretraining is followed by either: (i) direct fine-tuning on labeled data, or (ii) joint fine-tuning, where we optimize a combined objective that includes both classification and MLM losses—a strategy introduced in this work. Lastly, we evaluate our full model: Supervised Contrastive-Masked Pretraining (SCMP), which extends CMP by incorporating a supervised contrastive loss during pretraining, followed by joint fine-tuning. This setup allows us to isolate the impact of domain adaptation, joint objective optimization, and SCMP.

As shown in Figure 4, each step in our framework contributes to measurable performance improvements. Starting with CMP + Direct Fine-Tuning, we achieve a weighted F1-score of 0.83 and an accuracy of

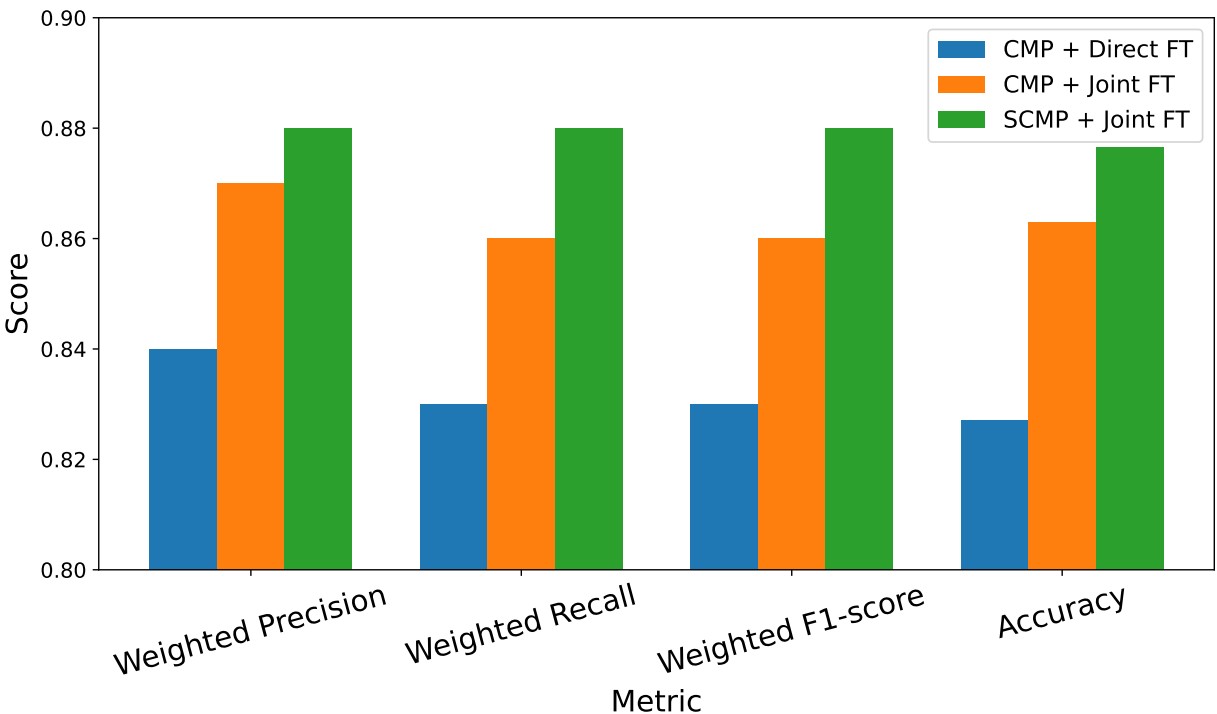

Figure 4: Comparison of weighted performance metrics across three training strategies. CMP = Continued Masked Pretraining; SCMP = Supervised Contrastive-Masked Pretraining; FT = Fine-Tuning.

0.8270. Adding joint fine-tuning leads to a consistent boost across all metrics, with weighted precision, recall, and F1-score each rising to 0.86, and accuracy improving to 0.8630. Incorporating SCMP with joint fine tuning yields the best results, with all weighted metrics reaching 0.88 and accuracy further improving to 0.8765. These results demonstrate that incorporating Supervised Contrastive-Masked Pretraining (SCMP), which combines supervised contrastive learning with masked language modeling, together with joint fine-tuning—optimizing classification and MLM objectives simultaneously—substantially enhances performance in low-resource classification tasks, achieving consistent improvements across all evaluation metrics.

## 4    Conclusion

In this paper, we introduced *Supervised Contrastive-Masked Pretraining (SCMP)*, a novel framework that integrates supervised contrastive learning with masked language modeling (MLM) to address the challenges inherent in low-resource language classification. By simultaneously optimizing for global semantic relationships through contrastive loss and local contextual dependencies via MLM, SCMP effectively enhances feature discrimination and preserves essential linguistic nuances. To facilitate efficient training, we proposed a *generalized batch processing framework for supervised contrastive loss*, as formulated in Equation 5, enabling scalable and stable optimization across varying batch sizes. Furthermore, we introduced a *fine-tuning loss function* that jointly optimizes MLM and classification loss, reinforcing contextual understanding while improving classification accuracy. Our experiments on sixteen languages from the *IndicNLP News Article Classification Dataset* (Kunchukuttan et al., 2020) and the *MasakhaNEWS* dataset (Adelani et al., 2023) demonstrate that SCMP with joint fine-tuning consistently outperforms the traditional fine-tuning approach, as evidenced by improved classification metrics..

The results validate that leveraging supervised contrastive objectives in conjunction with MLM not only refines the learned representations but also enhances the model's ability to capture language-specific semantics, an essential factor in low-resource language classification. Additionally, our proposed batch processing

framework ensures efficient contrastive learning, making it applicable to diverse NLP settings. While our study focuses on news classification, the promising outcomes of SCMP suggest its potential applicability to a broader range of low-resource languages and NLP tasks.

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

# Appendix

## A  Training Hyperparameters and Implementation Details

In this section, we provide detailed information about our experimental setup, including the hard-ware/software environment, training hyperparameters, and key implementation details for both the pre-training and fine-tuning stages. All experiments were conducted using NVIDIA Tesla A100 GPUs (80 GB) with Python 3.12.2.

### A.1  Pretraining: Supervised Contrastive-Masked Pretraining (SCMP)

In the pretraining stage, our goal was to jointly optimize a supervised contrastive loss with a masked language modeling (MLM) loss. Key implementation details include:

- **Data Sampling:** For each iteration, we randomly permuted the class labels and selected two random samples per class from the training set. This ensured that each mini-batch contained paired samples from every class.

- **Tokenization and Masking:** The training texts were tokenized with a maximum sequence length of 512. A masking probability of 30% was applied to non-padding tokens, and unmasked tokens were set to `-100` in the label tensor to ignore them during loss computation.

- **Optimization:** We employed the Adam optimizer with a learning rate of $5 \times 10^{-5}$ for pretraining. With each iteration, the loss was accumulated over four batch processing steps before performing an optimization update. In each batch processing step, the contrastive loss (Equation 5) was computed, while the MLM loss was directly obtained through the Hugging Face Transformers Model API. Pretraining was continued for 4000 iterations for each language with the same set of hyperparameters.

### A.2  Fine-Tuning

After pretraining, we fine-tuned the model for classification by adding a linear head on top of the pretrained language model. The classifier leverages the pretrained masked language model, loaded from a checkpoint, with a linear layer to obtain class logits from the CLS (Devlin, 2018) embedding. The total loss combines cross-entropy loss for classification and MLM loss with 30% masking. Fine-tuning was conducted for 50 epochs using the Adam optimizer with a learning rate of $5 \times 10^{-5}$, with a minibatch size of *B=16*. We used the same set of hyperparameters for all the languages.

## B  Comparison with Contrastive and Domain-Adaptive Baselines

To provide a thorough comparison of SCMP with existing contrastive and multilingual representation learn-ing techniques, we evaluate our method against three major families of baselines: (i) contrastive sentence representation learning, using **SimCSE** (Gao et al., 2021); (ii) multilingual contrastive encoders, represented by **LaBSE** (Feng et al., 2022); and (iii) **domain-adaptive pretraining (DAPT)** approaches that con-tinue masked language model pretraining on in-domain corpora (Gururangan et al., 2020). For fairness, both **SimCSE** and **DAPT** use the same *DistilBERT-multilingual-cased* backbone as SCMP, ensuring that perfor-mance differences stem from the training objective rather than model capacity. All baselines are trained and evaluated under identical data splits. Table 4 reports weighted F1-scores for all sixteen languages averaged over five runs.

Table 4: Language-wise weighted F1-scores comparing SCMP + Joint Fine-Tuning with SimCSE, LaBSE, and domain-adaptive pretraining (DAPT). Best score per language is shown in bold; ties are bolded.

| Language | SimCSE | LaBSE | DAPT | SCMP + Joint FT (ours) |
|---|---|---|---|---|
| *Indic languages* | | | | |
| Bengali | 0.95 | **0.98** | 0.95 | **0.98** |
| Gujarati | 0.76 | 0.99 | 0.94 | **0.99** |
| Kannada | 0.93 | 0.98 | 0.95 | **0.99** |
| Malayalam | 0.73 | 0.91 | 0.87 | **0.94** |
| Marathi | 0.89 | **0.99** | 0.94 | **0.99** |
| Oriya | 0.64 | **0.97** | 0.68 | 0.74 |
| Punjabi | 0.82 | 0.97 | 0.93 | **0.98** |
| Tamil | 0.90 | 0.97 | 0.93 | **0.98** |
| Telugu | 0.92 | **0.99** | 0.95 | **0.99** |
| *African languages* | | | | |
| Hausa | 0.60 | 0.83 | 0.80 | **0.86** |
| Igbo | 0.60 | 0.74 | 0.70 | **0.83** |
| Kiswahili | 0.63 | 0.78 | 0.72 | **0.79** |
| Luganda | 0.43 | 0.47 | 0.77 | **0.86** |
| Lingala | 0.63 | 0.72 | 0.83 | **0.89** |
| Pidgin | 0.81 | 0.87 | 0.94 | **0.98** |
| Yorùbá | 0.60 | 0.83 | 0.82 | **0.89** |

**Summary.** Across the sixteen languages, we observe clear performance differences among the baselines. **SimCSE** consistently performs the weakest—particularly in low-resource African languages (0.43–0.81)—likely because its sentence-level contrastive objective depends on abundant, diverse training data to form robust semantic anchors; in low-resource settings, this leads to unstable representations that fail to capture the fine-grained, token-level distinctions needed for news classification. **LaBSE** achieves strong results in higher-resource Indic languages (0.97–0.99) but shows less consistent behavior in African languages. **DAPT** provides gains, especially for Luganda and Lingala, but lacks an explicit mechanism to enforce class separation and therefore remains inconsistent across languages. In contrast, **SCMP + Joint Fine-Tuning** delivers consistent performance, achieving 0.94–0.99 across Indic languages and strong improvements in all African languages. These results demonstrate that combining supervised contrastive learning with masked language modeling produces more stable, discriminative representations, offering a robust solution for low-resource multilingual news classification.

## C   Computational Efficiency Analysis

To assess the computational efficiency of the proposed batch processing formulation used in SCMP pretraining, we conducted a controlled comparison against a baseline contrastive loss implementation that computes pairwise distances without tensorized operations. Both methods were evaluated under identical conditions using the same encoder (DistilBERT-multilingual), batch construction, optimizer settings, and GPU hardware. We used PyTorch's built-in timing and memory utilities to measure wall-clock time, throughput, and peak GPU memory usage.

The following metrics were recorded:

- **Time per epoch (s):** wall-clock time required for a full epoch.

- **Throughput (samples/s):** number of samples processed per second.

- **Mean peak GPU memory (GB):** maximum GPU memory allocated.

Table 5: Comparison of computational efficiency between the proposed batch processing formulation and the baseline untensorized contrastive loss implementation used during pretraining. Metrics include time per epoch, training throughput, and peak GPU memory usage.

| Metric | Batch Processing (Proposed) | Baseline Contrastive Loss (Untensorized) |
|---|---|---|
| Time per epoch (s) | $0.84 \pm 0.10$ | $0.86 \pm 0.07$ |
| Throughput (samples/s) | $38.6 \pm 5.9$ | $37.4 \pm 3.4$ |
| Mean peak GPU memory (GB) | 18.28 | 18.28 |

Table 5 summarizes the results. The proposed tensorized batch processing formulation achieves a slightly lower time per epoch ($0.84\,\mathrm{s}$ vs. $0.86\,\mathrm{s}$) and marginally higher throughput (38.6 vs. 37.4 samples/s) compared to the untensorized baseline, while both approaches require identical peak GPU memory. Although the absolute runtime difference is small at this scale, the tensorized formulation keeps all $n \times n$ pairwise computations within optimized CUDA kernels, avoiding Python-level loops and thereby offering better scalability as $n$ increases.

Overall, these results demonstrate that the tensorized formulation introduces no additional computational or memory overhead while enabling a more scalable implementation of supervised contrastive pretraining.

## D  Deriving the Supervised Contrastive Component of $\mathcal{L}_{\mathrm{CML}}$

The contrastive component of $\mathcal{L}_{\mathrm{CML}}$ in Eq. equation 3 is a direct specialization of the supervised contrastive (SupCon) loss of Khosla et al. (2020). The SupCon loss for an anchor embedding $z_i$ with positive set $P(i)$ is

$$\mathcal{L}_i^{\mathrm{SupCon}} = -\frac{1}{|P(i)|} \sum_{p \in P(i)} \log \frac{\exp(\mathrm{sim}(z_i, z_p)/\tau)}{\sum_{a \in A(i)} \exp(\mathrm{sim}(z_i, z_a)/\tau)}. \tag{7}$$

In our setting, each class $c \in \{1, \dots, n\}$ contributes two masked samples $\tilde{\mathbf{x}}_{1,c}$ and $\tilde{\mathbf{x}}_{2,c}$ with embeddings $h(\tilde{\mathbf{x}}_{1,c})$ and $h(\tilde{\mathbf{x}}_{2,c})$. For anchor $h(\tilde{\mathbf{x}}_{1,c})$ the only positive is $h(\tilde{\mathbf{x}}_{2,c})$, while all $h(\tilde{\mathbf{x}}_{2,c'})$ for $c' \neq c$ serve as negatives, so $|P(c)| = 1$. We use the negative squared Euclidean distance as the similarity measure:

$$\mathrm{sim}\big(h(\tilde{\mathbf{x}}_{1,c}), h(\tilde{\mathbf{x}}_{2,c'})\big) = -\|h(\tilde{\mathbf{x}}_{1,c}) - h(\tilde{\mathbf{x}}_{2,c'})\|^2, \tag{8}$$

with the temperature $\tau$ absorbed into the scaling.

Substituting equation 8 into equation 7 yields the numerator

$$\exp\big(-\|h(\tilde{\mathbf{x}}_{1,c}) - h(\tilde{\mathbf{x}}_{2,c})\|^2\big),$$

and the denominator

$$\sum_{c'=1}^{n} \exp\big(-\|h(\tilde{\mathbf{x}}_{1,c}) - h(\tilde{\mathbf{x}}_{2,c'})\|^2\big).$$

Since $|P(c)| = 1$, the per-class SupCon loss becomes

$$\mathcal{L}_c^{\mathrm{SupCon}} = -\log \frac{\exp\big(-\|h(\tilde{\mathbf{x}}_{1,c}) - h(\tilde{\mathbf{x}}_{2,c})\|^2\big)}{\sum_{c'=1}^{n} \exp(-\|h(\tilde{\mathbf{x}}_{1,c}) - h(\tilde{\mathbf{x}}_{2,c'})\|^2)}. \tag{9}$$

Summing equation 9 over $c = 1, \dots, n$ gives

$$-\sum_{c=1}^{n} \log \frac{\exp\big(-\|h(\tilde{\mathbf{x}}_{1,c}) - h(\tilde{\mathbf{x}}_{2,c})\|^2\big)}{\sum_{c'=1}^{n} \exp(-\|h(\tilde{\mathbf{x}}_{1,c}) - h(\tilde{\mathbf{x}}_{2,c'})\|^2)},$$

which matches exactly the contrastive term in Eq. equation 3. This shows that the contrastive part of $\mathcal{L}_{\mathrm{CML}}$ is precisely the supervised contrastive loss applied in the special case where each anchor has one positive (its class-paired sample) and all remaining classes act as negatives.

