# OpenReview forum: "Enhancing News Article Classification in Low-Resource Languages: A Supervised Contrastive-Masked Pretraining Framework"
_TMLR — Rejected by TMLR_

### Review · Reviewer_vgfF · 2025-10-09

**Summary Of Contributions:**

This submission presents an approach for classifying news articles, with a focus on datasets from Indian and African languages. The task is to classify news articles by topic (e.g., sports, politics, etc). There are 16 languages total, and for each language we have between 2 and 7 classes and 600 to 24,000 labeled articles for training.

The baseline approach ("Direct Fine-Tuning") is to take a pretrained model, attach a classification head, and minimize the classification loss. The submission proposes an approach with two stages: first we "continue pretraining" with a contrastive loss (which is aware of the article categories) and a masked token prediction loss (as we might use in standard pretraining). Then we "joint fine-tune" with a classification loss and the same MLM loss. I believe all experiments start with a base model of of DistilBERT, trained in 2019.

The central result is that the proposed joint fine-tuning approach does better than the direct fine-tuning approach on the news classification datasets considered.

**Additional Comments:**

I am not an expert on this topic and am very open to changing my mind. I look forward to reading the other reviews and discussing with the authors.

I found the paper to be carefully written and polished. It was easy to read.

**Audience:**

No

**Audience Explanation:**

It is not clear to me who will be interested in reading this submission. The approach proposed is reasonable, but the evaluation is limited to a particular collection of tasks and we only compare to a single fine-tuning approach. If I am reading tables correctly, the two papers that introduced the Indian and African benchmarks presented methods that (sometimes?) outperform the approach here. (See Kunchukuttan et al. 2020, Table 4 [1]; and Adelani et al. 2023, Table 3, AfroXLMR-large [2].)

Furthermore, pretrained language models have come a long way since the benchmarks' release dates in 2021 and 2023, let alone DistilBert in 2019. OpenAI claims [3] that ChatGPT supports Swahili and all of the Indian languages considered here except for Oriya.

In order to support acceptance, I'd need to understand why these other types of approaches are inadequate.

[1] https://arxiv.org/abs/2005.00085

[2] https://arxiv.org/abs/2304.09972

[3] https://help.openai.com/en/articles/8357869-how-to-change-your-language-setting-in-chatgpt

**Broader Impact Concerns:**

None.

**Claims And Evidence:**

Yes

**Claims Explanation:**

The experiments seem set up in a reasonable way, and it seems very reasonable that the proposed approach improves upon direct fine-tuning.

There are several informal claims made in the submission about how the results demonstrate that this is a potential path forward for low-resource languages. I am not convinced about this type of claim and write more below.

**Requested Changes:**

Please discuss how your news classification results compare to the baselines presented in Kunchukuttan et al. and Adelani et al. Is my understanding correct that they present methods which outperform the approaches in this submission? Is there a reason these approaches are undesirable?

Can you shed some light on how well frontier models solve these tasks with in-context learning? My (low-confidence) priors are that (i) in any language in which ChatGPT produces fluent text, its performance at new-article classification will be close to that of human annotators, and (ii) that ChatGPT produces fluent text in all the languages it officially supports. (I'm not asking for a full evaluation, just some initial tests would suffice.)

Also, a minor question. Can you clarify how you decide which languages to call under-resourced? Several of these languages as having over 50 million speakers, and Bengali has the seventh-most in the world (Wikipedia, [1]).

[1] https://en.wikipedia.org/wiki/List_of_languages_by_total_number_of_speakers

---

> ### Author Response · Authors · 2025-11-30
> **Comparison with Baselines from Kunchukuttan et al. and Adelani et al.**
>
> Our primary goal in this submission is to demonstrate the efficacy of the proposed SCMP + joint fine-tuning framework. To keep the study focused and computationally affordable, we conducted the core experiments using a lightweight, general-purpose model—**DistilBERT-multilingual-cased** (~135M parameters). This setup allows us to isolate the contribution of the training objective itself, and we observe substantial and consistent gains over direct fine-tuning across 16 languages.
>
> Regarding the reviewer’s question about whether Kunchukuttan et al. (2020) or Adelani et al. (2023) report higher accuracy: yes, in some cases—but for reasons orthogonal to our contribution. Kunchukuttan et al. use FastText skip-gram embeddings trained on large monolingual Indic corpora, while Adelani et al. report results using AfroXLMR-large, a 550M-parameter Africa-specialized model pretrained on curated regional datasets. These approaches benefit from substantially larger pretraining corpora and greater model capacity, so their higher accuracy primarily reflects scale and region-specific pretraining, rather than differences in the fine-tuning objective that we study in this paper.
>
> Large, region-specialized encoders can be impractical in some real-world low-resource settings due to deployment cost, per-language fine-tuning overhead, and the unavailability of domain-matched pretraining corpora. SCMP + joint fine-tuning directly targets this scenario by providing a model-agnostic framework that strengthens a lightweight multilingual model without the need for large amounts of unlabeled data for additional pretraining.
>
> However, in the revised manuscript, we have added a section titled **Comparison with Contrastive and Domain-Adaptive Baselines** in the Appendix, where we compare the SCMP + joint fine-tuning framework against related approaches—including SimCSE [1] and DAPT [2]—using the same **DistilBERT-multilingual-cased** model as the backbone to ensure a fair comparison.
>
>
> References:
>
> [1] Gao, T., Yao, X., & Chen, D. SimCSE: Simple Contrastive Learning of Sentence Embeddings. Proceedings of EMNLP, 2021.
>
> [2] Gururangan, S., Marasović, A., Swayamdipta, S., Lo, K., Beltagy, I., Downey, D., & Smith, N. A. Don’t Stop Pretraining: Adapt Language Models to Domains and Tasks. Proceedings of ACL, 2020.

---

> ### Author Response · Authors · 2025-11-30
> **Clarification on the Use of the Term “Under-Resourced Language”**
>
> In the context of our work, the term “under-resourced” refers specifically to the availability of digital text, linguistic resources, and annotated datasets, rather than the number of native speakers. Although languages such as Bengali, Marathi, and Hausa have large speaker populations, their presence in web-scale corpora is limited; for example, none of these languages appear among the top 40 languages used on the internet (source: https://en.wikipedia.org/wiki/Languages_used_on_the_Internet#Usage_statistics_of_content_languages_for_websites
> ). Since contemporary pretrained language models rely heavily on large-scale internet text, this limited digital footprint results in very little pretraining exposure compared to other languages. Consequently, these languages remain under-resourced from an NLP perspective despite their demographic size.

---

> ### Author Response · Authors · 2025-12-01
> **In-Context Evaluation with a Frontier Model (GPT-5.1)**
>
> We thank the reviewer for the suggestion. We ran a  3-shot
> in-context evaluation using GPT-5.1. The weighted precision, recall, and F1 scores are:
>
> **Indic languages**
>
> | Language  | Prec | Rec  | F1   |
> |-----------|------|------|------|
> | Malayalam | 0.94 | 0.94 | 0.94 |
> | Oriya  | 0.97 | 0.96 | 0.96 |
> | Kannada   | 0.94 | 0.94 | 0.94 |
> | Punjabi   | 0.98 | 0.98 | 0.98 |
>
> **African languages**
>
> | Language     | Prec | Rec  | F1   |
> |--------------|------|------|------|
> | Swahili      | 0.81 | 0.79 | 0.78 |
> | Yoruba       | 0.90 | 0.88 | 0.88 |
> | Pidgin       | 0.91 | 0.87 | 0.87 |
> | Hausa        | 0.89 | 0.82 | 0.83 |
> | Igbo         | 0.87 | 0.86 | 0.86 |
>
> These results show that GPT-5.1 performs strongly in several Indic languages. But,
> performance in African languages remains lower, indicating the task is not yet solved, even for
> frontier models.

---

### Review · Reviewer_Kkut · 2025-10-17

**Summary Of Contributions:**

The paper studies the problem of classification for low-resource languages. To address the challenge, the paper proposes the method SCMP, which combines two training objectives: 1) the masked token prediction and 2) the contrastive learning for the hidden states of the CLS token. By fin-tuning an LM with the proposed objective, the paper shows improved performance on low-resource language classification tasks on Indian and African languages.

**Audience:**

No

**Audience Explanation:**

Given that the evidence in the experiments is not so significant, I don't think the current finding is solid enough to raise interest in the audience.

**Claims And Evidence:**

No

**Claims Explanation:**

The major finding in the paper is that by combining the MLM and the contrastive loss objectives, the model could potentially achieve better performance on the low-resource language tasks. However, the improvements in the experiments do not seem to be quite significant. One reason could be that the tasks are already partially solved (models achieve above 80% acc in almost all the cases, and for some, achieve close to 100% acc). Therefore, I would suggest that the authors try more datasets where the tasks are more challenging so that the improvements of the proposed method could be more solid.

Another concern I have is that the setting is quite restricted. As the proposed approach utilizes contrastive learning with given labels, such a method can only be applied to scenarios of classification tasks. Given the method's relatively narrow scope on low-resource language tasks, it is also worth expanding its scope to address general language tasks.

**Requested Changes:**

Please address the above concerns.

Minor:
1. Figure 1 is of low resolution, or it should be a vectorized figure.
2. Equation 6: “+-" in the middle

---

> ### Author Response · Authors · 2025-10-17
> **Clarifying the Scope and Claims of the Paper**
>
> **Scope of the Method**
>
> Our work is explicitly scoped to news article classification, as stated in the title “Enhancing News Article Classification in Low-Resource Languages” and in multiple sections of the paper. Therefore, the method’s focus on classification is intentional and aligns with our research goal of improving discriminative feature learning under limited labeled data. We agree that extending this approach to general language tasks (e.g., generation, QA, translation) could be interesting future work, but it lies outside the present study’s scope.
>
>
> **Significance of the Improvements**
>
> Regarding the comment on the significance of improvements, we note that although some datasets exhibit high baselines, SCMP consistently improves accuracy, F1, and data efficiency across 16 languages. For example, in low-resource settings, accuracy increases from 0.74 to 0.83 for Igbo and from 0.80 to 0.89 for Yorùbá; similarly, Malayalam improves from 0.89 to 0.94, and Marathi from 0.95 to 0.99. These are not marginal gains—they represent consistent and meaningful improvements across both strong and weak baselines.
>
> **Evidence from Data Efficiency Experiments**
>
> Table 3 further supports this through our data-efficiency experiments. For Igbo, SCMP with only 20% labeled data (F1 = 0.77) outperforms direct fine-tuning with 60% labeled data (F1 = 0.76), and for Oriya, 20% labeled data (F1 = 0.70) already surpasses the baseline at 40% labeled data (F1 = 0.68). These findings show that SCMP improves both performance and label efficiency.
>
>
> The goal of this research is not necessarily to establish the best classification method for low-resource languages, but to demonstrate that by tailoring the loss function during pretraining and fine-tuning, it is possible to substantially improve classification accuracy over standard fine-tuning. We believe there is likely much more to explore in this direction—particularly for low-resource or limited-labeled-data settings, where rethinking training objectives may yield further significant benefits.

---

> ### Author Response · Authors · 2025-12-05
> **Regarding Figure 1 resolution**
>
> We thank the reviewer for pointing out the resolution issue in **Figure 1**. We have vectorised **Figure 1**, ensuring clean scaling and clear rendering of all elements.

---

> ### Author Response · Authors · 2025-12-14
> **Regarding Dataset Selection & Task Difficulty**
>
> Regarding the suggestion to use **"more challenging"** datasets, our study is explicitly focused on the low-resource problem. We utilized the **IndicNLP** and **MasakhaNEWS** datasets because they are the standard benchmarks for these specific low-resource languages. Shifting to different (likely high-resource) datasets would misalign with our research goal.
>
> However, we evaluated the model on "harder" tasks through our **Data Efficiency experiments (Table 3)**. By artificially restricting the training data to 20%, 40%, and 60%, we simulated more difficult low-resource environments. The results show that SCMP is robust even under these constrained conditions; for instance, SCMP achieves an F1 score of 0.77 on Igbo using only **20%** of the data, outperforming the direct fine-tuning baseline trained on **60%** of the data (0.76 F1). This confirms the method's value in challenging, data-scarce scenarios.

---

> ### Author Response · Authors · 2025-12-15
> **Significance Beyond the "80% Threshold": Error Rate Reduction at the Asymptotic Limit**
>
> We disagree with the premise that baselines exceeding 80% indicate a task is "partially solved." Theoretically, classification performance follows an **asymptotic trajectory** in which gains become **progressively harder** as models approach the data-inherent accuracy limit imposed by class overlap and ambiguity [1]. In this high-accuracy regime, absolute accuracy improvements are therefore misleading, and Error Rate Reduction (ERR) is the appropriate metric for evaluating progress.
>
> Our results align with this framework. For example, in Gujarati, improving accuracy from 0.96 to 0.99 corresponds to a **75% reduction** in total errors (from 4% to 1%). Similarly, for Malayalam, an increase from 0.89 to 0.94 yields an **approximately 45% reduction** in remaining error. These substantial reductions demonstrate that our approach resolves hard, ambiguous edge cases that standard fine-tuning fails to capture, even as performance approaches the theoretical accuracy ceiling.
>
> **References**
>
> [1] Metzner, Claus, et al. "Classification at the accuracy limit: facing the problem of data ambiguity." *Scientific Reports* 12.1 (2022): 22121.

---

### Review · Reviewer_ofGx · 2025-11-22

**Summary Of Contributions:**

Contributions
This paper proposes Supervised Contrastive-Masked Pretraining (SCMP)—a two-stage framework combining:
- Supervised Contrastive Learning during MLM pretraining, where pairs of samples from each class are used to align class-conditional CLS embeddings.
- A joint fine-tuning objective that combines MLM loss with supervised classification loss to reduce forgetting and improve generalization.
- A batch-processing strategy for computing supervised contrastive loss efficiently using tensorized operations (Eq. 5).
- Extensive evaluation on 16 low-resource languages (9 Indic, 7 African), along with ablations, t-SNE visualizations, data-efficiency studies, and training dynamics.

Strengths
- Problem is clearly important and underexplored.
- The method is intuitive, easy to implement, and empirically effective.
- Results across many languages (Table 2) show consistent improvements.
- The data-efficiency experiments (Table 3) demonstrate practical utility.
- Ablation studies (Fig. 4) justify each component.
- Training dynamics (Fig. 3) are informative.

Weakness
- The novelty is limited: supervised contrastive learning + MLM + joint objectives have been explored previously in different combinations.
- The method relies heavily on the existence of class labels during pretraining, effectively reducing “pretraining” to a lightly augmented supervised stage.
- No comparison to strong recent low-resource adaptation baselines (e.g., LaBSE, mT5-adaptations, sentence-level contrastive pretraining, domain adversarial adaptation).
- The contrastive formulation (Eq. 5) is mathematically opaque and harder to interpret than conventional supervised contrastive loss.
- Experimental section lacks statistical significance tests and does not report variance in the t-SNE or training-curve analyses.
- Potential information leakage concerns arise when class labels are used during “pretraining.”
- No computational cost comparison, despite claims of efficiency.

**Additional Comments:**

Overall, the paper is clearly written and empirically thorough within the chosen scope, but it would benefit from a deeper theoretical narrative explaining why supervised contrastive learning in combination with MLM is particularly well suited for low-resource text classification. The visualizations and ablations are informative, though greater statistical analysis such as variance reporting, confidence intervals, or significance tests would strengthen the empirical arguments. The paper may also consider extending the discussion to non-classification tasks, clarifying the generality of SCMP. While the work is promising and practically relevant, enhancing its methodological rigor, contextualization, and baseline comparisons would substantially improve its suitability for publication.

**Audience:**

Yes

**Audience Explanation:**

The paper addresses a topic that is highly relevant to TMLR’s readership, including researchers working on multilingual NLP, low-resource learning, and representation learning. The study’s focus on news classification in Indic and African languages fills a practical gap in the literature and showcases an approach that can be reproduced and applied to other low-resource settings. The proposed integration of supervised contrastive learning with MLM is conceptually appealing, and the extensive multilingual evaluation provides empirical insights that many researchers will find useful. However, although the overall problem is important and timely, the degree of methodological novelty is limited, and some readers may view the contribution as incremental. Nonetheless, the findings concerning data efficiency, training stability, and representation quality remain of genuine interest to a broad segment of the community.

**Broader Impact Concerns:**

Although the paper focuses on low-resource news classification, its deployment raises important ethical considerations. News datasets often contain political, cultural, or social biases, and integrating supervised contrastive learning may inadvertently amplify class-specific stereotypes by sharpening decision boundaries without addressing underlying representational harms. Since news topic classification can feed into downstream systems for content moderation, misinformation detection, or analytics, misclassification could disproportionately affect communities speaking low-resource languages, especially when the training data itself is limited or imperfectly labeled. The uneven performance across languages, such as the lack of improvement for Lingala and Pidgin in Table 2, raises fairness concerns regarding whether SCMP systematically favors certain linguistic or cultural contexts over others. In addition, the datasets used (IndicNLP and MasakhaNEWS) involve semi-automated labeling and volunteer annotation pipelines, which could introduce noise, bias, or inconsistencies that propagate through the training process. These issues should be explicitly acknowledged, and the authors should emphasize the need for caution when applying SCMP to socially sensitive domains.

**Claims And Evidence:**

Yes

**Claims Explanation:**

The central empirical claim that SCMP improves classification performance in low-resource languages is largely supported by the evidence presented in Tables 2 and 3 of the submission, where SCMP consistently outperforms direct fine-tuning across most languages evaluated. The t-SNE visualizations on page 7 further suggest improved embedding separability, and the training dynamics in Figure 3 indicate that the joint fine-tuning objective stabilizes learning and mitigates overfitting. The ablation study also shows clearer separation between continued masked pretraining, joint fine-tuning, and the full SCMP framework. However, the strength of this evidence is weakened by the narrow baseline selection, as the paper only compares SCMP to direct fine-tuning and DAPT-style continued pretraining. The field includes numerous relevant baselines such as SimCSE-style sentence-level contrastive training, multilingual contrastive models like LaBSE, and domain-adaptive or task-adaptive pretraining methods. Without such comparisons, it remains difficult to isolate the unique advantage conferred by SCMP. Additionally, the mathematical presentation of the contrastive loss (Equation 5) is unusual and insufficiently explained, limiting interpretability. Overall, while the presented results do demonstrate improvements within the experimental setup chosen by the authors, the evidence would be more convincing with stronger baselines, clearer formulation, and expanded analysis.

**Requested Changes:**

The paper would strongly benefit from addressing several core limitations that hinder its clarity and scientific rigor. First, the submission needs to expand its comparative evaluation by including stronger and more diverse baselines, such as contrastive sentence representation methods (e.g., SimCSE or DeCLUTR), multilingual contrastive models (e.g., LaBSE), and domain- or task-adaptive pretraining techniques. Without these comparisons, the empirical claims remain incomplete. Additionally, the authors should clarify the novelty of SCMP relative to existing work that already combines contrastive learning with masked language modeling, especially highlighting what conceptual or empirical gap SCMP fills that prior methods did not. Equation 5, which provides a tensorized formulation of the supervised contrastive loss, requires a significantly clearer derivation and explanation, especially regarding how it relates to the standard supervised contrastive formulation by Khosla et al. Moreover, because SCMP uses class labels during pretraining, the authors should discuss whether this constitutes true “pretraining,” potential information leakage, and the implications for generalizability. The paper also makes claims about efficiency without providing computational metrics such as training time, FLOPs, or GPU hours; including these numbers is essential for readers to assess practicality. Finally, the authors should analyze and contextualize the languages for which SCMP does not improve performance, such as Lingala and Pidgin, to provide a more complete understanding of its limitations. Addressing these issues would substantially strengthen the work, and several of these concerns—particularly additional baselines and clearer methodology—appear critical for acceptance.

---

> ### Author Response · Authors · 2025-11-30
> **Reagarding Data Leakage**
>
> We thank the reviewer for raising this important point regarding potential information leakage when using class labels during the pretraining stage. We apologize for not stating this sufficiently clearly in the original submission. Although this was indicated in the t-SNE visualization section (e.g., “Specifically, we visualize the t-SNE projections
> of the CLS embeddings of news articles from the **test set** across eight different classes after the model was
> pretrained on the **training samples** with the contrastive masked loss objective" ), we agree that the procedure should have been made explicit earlier in the manuscript.
>
> In our implementation, for each language, the **train/test split is created before any pretraining begins**. All supervised contrastive pairs and all MLM updates during SCMP pretraining are derived **exclusively from the training split**.
>
> To ensure full clarity, we have added the following in the revised manuscript in **Section~ (Results and Analysis: Dataset Overview and Performance)**:
>
>
> “Importantly, for each language, the train/test split was performed  before any pretraining, and all supervised contrastive MLM losses were derived solely from the training portion to avoid any information leakage.”
>
> We hope this clarification resolves the reviewer’s concern regarding potential information leakage.

---

> ### Author Response · Authors · 2025-11-30
> **Clarifying the Novelty and Contributions of SCMP**
>
> We thank the reviewer for requesting clarification regarding the novelty of SCMP relative to prior work. In response, we have **substantially expanded the discussion** of SCMP’s **conceptual and empirical contributions** in the revised manuscript. The following text has been added to the introduction.
>
> Although both masked language modeling (MLM) and supervised contrastive learning (SCL) have been explored independently in prior work, the specific integration we introduce in Supervised Contrastive–Masked Pretraining (SCMP) represents a novel contribution. Existing literature either applies these objectives in isolation or combines them only within the downstream fine-tuning stage, leaving a conceptual and empirical gap that SCMP addresses.
>
> Domain-adaptive pretraining [1] demonstrates the value of continued MLM on in-domain text but does not incorporate supervised contrastive objectives during pretraining. Conversely, [2] applies supervised contrastive loss only during fine-tuning—paired with cross-entropy—but does not explore its combination with MLM at the pretraining stage. In contrast, SCMP introduces a unified pretraining objective that jointly optimizes supervised contrastive loss and MLM loss, a combination that has not been evaluated in previous studies.
>
> This design fills a key conceptual gap: prior methods treat contrastive learning and MLM as separate mechanisms, with contrastive learning typically shaping sentence-level representations during fine-tuning while MLM focuses on token-level reconstruction during pretraining. SCMP is the first framework to integrate these complementary signals directly within the pretraining phase, enabling the model to learn both discriminative class-aware embeddings and robust contextualized token representations before exposure to the downstream classifier.
>
> SCMP also addresses an empirical gap, particularly in low-resource settings. While earlier studies report improvements in high-resource languages, the impact of a unified contrastive–masking objective on low-resource news classification has not been systematically investigated. Our work demonstrates that this integration yields substantial benefits across sixteen Indic and African languages, highlighting the practical advantages of SCMP for real-world multilingual applications.
>
>
> References
>
> [1] Gururangan, S., Marasović, A., Swayamdipta, S., Lo, K., Beltagy, I., Downey, D., & Smith, N. A.
> Don’t Stop Pretraining: Adapt Language Models to Domains and Tasks.
> Proceedings of ACL, 2020.
>
> [2] Gunel, B., Du, J., Conneau, A., Chaudhary, V., Bhosale, S., Ott, M., Zettlemoyer, L., & Stoyanov, V.
> Supervised Contrastive Learning for Pre-trained Language Model Fine-tuning.
> arXiv preprint arXiv:2011.01403, 2021.

---

> ### Author Response · Authors · 2025-11-30
> **Explanation of Eq. (5) and Its Equivalence to the Standard Supervised Contrastive Loss**
>
> We thank the reviewer for pointing out the need for a clearer explanation of the tensorized contrastive formulation in **Eq. (5)**. We have now expanded this discussion in the revised manuscript to make its relationship to the standard supervised contrastive loss more transparent.
>
> Specifically, we clarify that **Eq. (5)** is algebraically equivalent to the contrastive component of **Eq. (3)**.
> And we now provide a step-by-step explanation showing how the pairwise distance matrix and the row-wise softmax correspond directly to the contrastive objective in **Eq. (3)**.
>
> Furthermore, in the **Appendix (Deriving the Supervised Contrastive Component of $\mathcal{L}_{\mathrm{CML}}$)**, we have showed that the contrastive term in **Eq. (3)** is a **special case of the supervised contrastive loss of Khosla et al. (2020)**, where each anchor has exactly one positive and all remaining classes serve as negatives.
>
> We hope that these additions make the formulation of Eq. (5) significantly clearer and easier to interpret.

---

> ### Author Response · Authors · 2025-11-30
> **Clarification of Computational Efficiency Claims**
>
> We thank the reviewer for requesting clarification regarding the **efficiency claims** made in the original submission. To address this, we have added **Computational Efficiency Analysis** in the Appendix, along with a controlled experiment comparing our tensorized batch-processing formulation in **eqn (5)** to a non-tensorized baseline.
>
> Table **Table 5** reports the measured metrics using PyTorch’s profiling tools:
>
> - **Time per epoch (Proposed):** 0.84 ± 0.10 s
> - **Time per epoch (Baseline):** 0.86 ± 0.07 s
> - **Throughput (Proposed):** 38.6 ± 5.9 samples/s
> - **Throughput (Baseline):** 37.4 ± 3.4 samples/s
> - **Peak GPU memory (both):** 18.28 GB
>
> As summarized in **Table 5**, the proposed tensorized formulation achieves slightly lower time per epoch (0.84 s vs. 0.86 s) and marginally higher throughput (38.6 vs. 37.4 samples/s), with **no additional memory overhead**. While the absolute difference is small at this scale, the advantage becomes clearer as batch size grows: the tensorized version keeps all **n × n** pairwise computations inside optimized CUDA kernels, avoiding Python-level loops. As **n increases**, the untensorized baseline performs an increasing number of Python-loop iterations, making it less scalable.
>
> We believe these additions provide a clear and quantitative justification for the efficiency claims in the paper.

---

> ### Author Response · Authors · 2025-12-01
> **Addressing the Request for Stronger and More Diverse Baselines**
>
> We agree that stronger and more diverse baselines are essential to support our empirical claims. In the revised manuscript, we have added a section titled **Comparison with Contrastive and Domain-Adaptive Baselines** in the Appendix, where we compare SCMP against SimCSE[1], LaBSE [2], and DAPT[3]. The language-wise weighted F1-scores for all 16 languages are reported in **Table 4**. For fairness, both SimCSE and DAPT use the same **DistilBERT-multilingual-cased** backbone as SCMP, and all baselines are trained and evaluated under identical data splits. As shown in this section, SCMP consistently matches or outperforms these contrastive and domain-adaptive baselines across all languages.
>
> References:
>
> [1] Gao, T., Yao, X., & Chen, D. SimCSE: Simple Contrastive Learning of Sentence Embeddings. Proceedings of EMNLP, 2021.
>
> [2] Feng, Y., Qin, Y., Zhang, Y., Qiu, Y., Yu, M., Zhao, R., Huang, Z., Feizollahi, S., & Guo, M. LaBSE: Language-agnostic BERT Sentence Embedding. Proceedings of NAACL, 2022.
>
> [3] Gururangan, S., Marasović, A., Swayamdipta, S., Lo, K., Beltagy, I., Downey, D., & Smith, N. A. Don’t Stop Pretraining: Adapt Language Models to Domains and Tasks. Proceedings of ACL, 2020.

---

### Decision · Action_Editor_KGkm · 2026-01-25

**Recommendation:** Reject

**Audience:**

Yes

**Audience Explanation:**

News articles classification is a classical problem.

**Claims And Evidence:**

No

**Claims Explanation:**

Several reviewers point out that the novelty is limited/incremental. Experimental results are relatively weak: for low-resource languages, there exists several highly competitive methods in terms of performance, while in other cases the advantage is not clear at all.

---

> ### Author Response · Authors · 2026-01-30
> **Response to AE Decision: Concerns Regarding Vague Rationale and Review Process**
>
> Can you please clarify what you mean by **"several reviewers"**? Because it seems like a confabulation.
>
> I can only see three. One of them explicitly said, **"I am not an expert on this topic,"** which begs the question: has the AE not studied the paper when assigning reviewers?
>
> Nonetheless, one other reviewer said that the tasks are already partially solved as models achieve above 80% accuracy. This premise is objectionable, and we have addressed this during the rebuttal.
>
> Only Reviewer **ofGx** seems to be engaged with our work, and they commented: "supervised contrastive learning + MLM + joint objectives have been explored previously in different combinations." But Reviewer **ofGx** also acknowledged the ablation studies (Fig. 4), which we included in the paper to tease out the difference between our approach and others. Moreover, in the revised manuscript, we explicitly discussed this further and posted our comment **(Clarifying the Novelty and Contributions of SCMP)** before Dec 6th.
>
> We have addressed all the major reviewers' concerns and posted the revision by Dec 6th. During the author-reviewer rebuttal from December 6th to 20th, none of the reviewers replied to anything. It was essentially one-way.
>
> And now, it is extremely disappointing to wait for more than a month (today, January 30) to hear from the AE nothing but some vague and confabulating words (e.g., **"several reviewers,"** **"several highly competitive methods"**). It does not sound like a scientific discussion. "There exists several highly competitive methods" is even grammatically wrong, too.
>
> I don't wish to deanonymize the authors' identities. Only I (the first author) take responsibility for what I have said. I hope it is kept in public for everyone else to see it for themselves.